# Synthesis and Antiproliferative Effect of 3,4,5-Trimethoxylated Chalcones on Colorectal and Prostatic Cancer Cells

**DOI:** 10.3390/ph17091207

**Published:** 2024-09-13

**Authors:** Cécile Letulle, François-Xavier Toublet, Aline Pinon, Soufyane Hba, Aurélie Laurent, Vincent Sol, Catherine Fagnère, Benjamin Rioux, Florent Allais, Sophie Michallet, Laurence Lafanechère, Youness Limami, Mounia Oudghiri, Mohamed Othman, Adam Daïch, Bertrand Liagre, Ata Martin Lawson, Christelle Pouget

**Affiliations:** 1Univ. Limoges, LABCiS, UR 22722, Faculty of Pharmacy, F-87000 Limoges, France; cecile.letulle@etu.univ-lehavre.fr (C.L.); francois-xavier.toublet@unilim.fr (F.-X.T.); aline.pinon@unilim.fr (A.P.); hbasoufyane@gmail.com (S.H.); aurelie.laurent@unilim.fr (A.L.); vincent.sol@unilim.fr (V.S.); catherine.fagnere@unilim.fr (C.F.); benjamin.rioux@cma-nouvelleaquitaine.fr (B.R.); bertrand.liagre@unilim.fr (B.L.); 2Université Le Havre Normandie, Normandie Univ, URCOM UR 3221, INC3M, FR CNRS 3038, 25 Rue Philippe Lebon, BP 1123, F-76063 Le Havre Cedex, France; mohamed.othman@univ-lehavre.fr (M.O.); adam.daich@univ-lehavre.fr (A.D.); 3Laboratory of Immunology and Biodiversity, Faculty of Sciences Ain Chock, Hassan II University, B.P. 2693, Maarif, Casablanca 20100, Morocco; youness.limami@gmail.com (Y.L.); mouniaoudghiri@gmail.com (M.O.); 4URD Agro-Biotechnologies Industrielles (ABI), CEBB (Centre Européen de Biotechnologie et de Bioéconomie), AgroParisTech, F-51110 Pomacle, France; florent.allais@agroparistech.fr; 5Univ. Grenoble Alpes, Institute for Advanced Biosciences, Team Cytoskeletal Dynamics and Nuclear Functions, INSERM U1209, CNRS UMR5309, F-38000 Grenoble, France; sophie.michallet@univ-grenoble-alpes.fr (S.M.); laurence.lafanechere@univ-grenoble-alpes.fr (L.L.); 6Laboratory of Health Sciences and Technologies, Higher Institute of Health Sciences, Hassan First University of Settat, Settat 26000, Morocco

**Keywords:** 3,4,5-trimethoxylated chalcones, antiproliferative activity, microtubule-depolymerizing agents, apoptosis induction, colorectal and prostatic cancer cells

## Abstract

In the context of designing innovative anticancer agents, the synthesis of a series of chalcones bearing a 3,4,5-trimethoxylated A ring and a variety of B rings, including phenols and original heterocycles such as chromones, was conducted. For this end, Claisen–Schmidt condensation was performed in basic or acidic conditions between the common starting material 3,4,5-trimethoxyacetophenone and appropriate aldehydes; this allowed the recovery of fifteen chalcones in moderate–good yields. The synthesized compounds were screened for their antiproliferative activity against colorectal and prostatic cancer cells, using a colorimetric MTT assay. Among the new chromonyl series, chalcone **13** demonstrates an interesting antiproliferative effect, with IC_50_ values in the range of 2.6–5.1 µM at 48 h. Then, our study evidenced that indolyl chalcone **10** exhibits excellent activity towards the selected cell lines (with IC_50_ less than 50 nM). This compound has already been described and has been shown to be a potent anticancer agent against other cancer cell lines. Our investigations highlighted apoptosis induction, through several pro-apoptotic markers, of these two heterocyclic chalcones. Considering phenolic chalcones, compounds **2** and **8** were found to be the most active against cell proliferation, exerting their effect by inducing the depolymerization of cell microtubules. The most promising compounds in this series will be selected for application in a strategy of vectorization by either active or passive targeting.

## 1. Introduction

Cancer is considered to be a major health issue since it is the second most recurrent death-causing factor after cardiovascular diseases. Estimates of cancer-related mortality made by GLOBOCAN 2020, produced by the International Agency for Research on Cancer, have shown approximately 19.3 million new cancer cases and almost 10.0 million cancer deaths worldwide in 2020 [1]. Colorectal cancer (CRC) appeared as the second cause of cancer death after lung cancer. In addition, prostate cancer (PCa) is the most commonly diagnosed cancer in men, though its mortality is lower than those of the cancers previously mentioned [1]. Therefore, we have long been interested in the development of new therapeutic agents directed against these two cancers [2,3,4]. Natural products have always been considered to be invaluable sources of inspiration for drug design. Thus, chalcones are polyphenolic compounds belonging to the vast family of flavonoids, widely distributed in the plant kingdom, found in fruits and vegetables [5,6]. Chalcones are open-chain molecules in which two aromatic rings are joined by a three-carbon α,β-unsaturated carbonyl system (Figure 1) [7]. They have received significant attention due to their wide range of biological activities, including antioxidant [8], anti-inflammatory [9], antibacterial [10], antiviral [11], and anticancer properties [7,12,13].

The in vitro antiproliferative activity of chalcones against cancer cell lines has been mostly reported and is associated with interfering with the activity of several mechanisms and targets, such as aromatase, VEGF, JAK/STAT signaling pathways, tubulin, cathepsin-K, and topoisomerase-II [14,15].

Considering the structure–activity relationship, a trimethoxyphenyl ring was demonstrated to be interesting for anticancer activity of chalcones [16,17]. Therefore, the present study reports the synthesis and antiproliferative activity of several chalcones bearing this trimethoxyphenyl A ring, while the B ring is found to either be substituted by hydroxy and/or methoxy groups or to consist of heterocyclic moieties.

First, some of our studies have pointed out the potential of the 3-hydroxy-3′,4,4′,5′-tetramethoxychalcone (named chalcone **8** in this paper) as an anticancer agent [2,4]. This chalcone was first reported by Ducki et al. [18]; it was developed as a combretastatin-A4 (**C-A4**) analogue (Figure 2) and found to be a potent inhibitor of tubulin polymerization. Our previous study also revealed that this chalcone inhibited the proliferation of HCT116 and HT-29 human CRC cells by inducing cell cycle arrest before triggering programmed cell death [2]. We demonstrated that apoptosis mainly occurs through the activation of the extrinsic apoptotic pathway. Consequently, some chalcones were designed with hydroxy and/or methoxy groups at different positions of the B ring to complete a structure–activity relationship study.

Then, the B ring was also modified, incorporating heterocyclic moieties; indeed, these rings are known to be bioactive and their incorporation on the chalcone scaffold was thought to further increase the antiproliferative activity. From this perspective, the indole cycle was selected since its derivatives have become an interesting anticancer drug scaffold due to their valuable pharmacokinetic and pharmacodynamic properties [19]. Thus, many inhibitors of tubulin polymerization based on the indolic structure have been reported [20,21]. In addition, four chalcones with a chromonyl scaffold were designed; such compounds have not yet been widely studied. Only a few molecules have been evaluated for antibacterial and antifungal activities, as mentioned by Siddiqui et al. [22], for example; however, chalcones associating a chromonyl B ring with a trimethoxyphenyl A ring are totally original. With this background, the present work led us to synthesize a series of fifteen chalcones and to evaluate their ability to inhibit the proliferation of two human CRC (HCT116 and HT-29) and two human PCa (DU145 and PC3) cell lines. Some of the lead molecules were further investigated for the elucidation of their mechanism of action: the ability of three chalcones to impact microtubule depolymerization was studied, as was the role of two other chalcones on apoptosis induction.

## 2. Results and Discussion

### 2.1. Chemistry

The protocol for the synthesis of chalcones is described in Figure 1, Figure 2, Figure 3 and Figure 4. All compounds were characterized by FT-IR, ^1^H NMR, ^13^C NMR, and mass spectral data. The ^1^H NMR, ^13^C NMR, and mass spectra of the lead compounds are presented in Appendix A (see Appendix A).

The synthetic pathway consists of a Claisen–Schmidt condensation between the 3,4,5-trimethoxyacetophenone which is commercially available and appropriate substituted aldehydes. For chalcones **1**–**10**, the aldehydes were also commercially accessible. The synthesis of chalcone **11** involved 6-oxo-1,6-dihydropyridine-3-carbaldehyde, which was obtained following the methodology described by Butora et al. (Appendix A) [23]. Concerning chalcones **12**–**15**, the 3-formylchromones were synthesized using an improved procedure described by Nohara et al. that exploited the Vilsmeier–Haack reagents (Appendix A) [24]. For whole compounds, the solvent used was ethanol. On the one hand, for chalcones **1**–**9** and **11**, the Claisen–Schmidt condensation was performed with classic conditions, i.e., using NaOH as the base (Figure 1 and Figure 2); for indolyl chalcone **10**, the base was piperidine (Figure 3), as previously described [25]. The synthesized chalcones were obtained as yellow powders, with yields in the range of 10–80%; for some chalcones, the lower yields were due to a decrease in the electrophilic character of the aldehydic carbonyl, which was related to the presence of the 2- or 4-hydroxy group, which was ionized in an alkaline medium. Therefore, for chalcones **3** and **9**, a protection step of the 4-hydroxy group of the benzaldehyde was performed using 3,4-dihydro-2H-pyran and pyridinium para-toluenesulfonate as catalyst (Figure 2). The corresponding tetrahydropyranyl ether was then reacted with 3,4,5-trimethoxyacetophenone, as previously described; finally, a deprotection step was carried out using para-toluenesulfonic acid in methanol at room temperature. Thus, chalcone **3** was obtained in a 68% yield (vs. 32% without protection), while chalcone **9** was synthesized in a 37% yield (vs. 12% without protection).

On the other hand, the synthesis of chromonyl chalcones **12**–**15** was achieved in acidic conditions (Figure 4), since an alkaline medium that was responsible for the degradation of the 3-formylchromone. p-Toluenesulfonic acid (PTSA, 12% in acetic acid) was selected as the catalyst; this enabled yields ranging from 60 to 100%.

The H-α and H-β of the whole chalcones both appeared as a doublet, with the coupling constant around 15.5 Hz, which agrees with the trans-configuration of α, β-unsaturated ketone.

### 2.2. Biological Activity

#### 2.2.1. Effect of Chalcones on Cell Viability

The resultant chalcones were tested for their in vitro antiproliferative activity by MTT assay (a metabolic activity test) against two human CRC (HCT116 and HT-29) and two human PCa (DU145 and PC3) cell lines.

IC_50_ values of the whole compounds in the MTT assay are presented in Table 1, Table 2, Table 3 and Table 4. Cell viability relative to concentration is represented in Figure 3, Figure 4, Figure 5, Figure 6 and Figure 7. For the two active chalcones **10** and **13**, the Trypan blue dye exclusion method was used to confirm the effect of our compounds on cell viability. The results of this additional test are reported in Table 5 and in the Appendix A (Appendix A). This Trypan blue test, showing IC_50_ very similar to that previously determined, allowed us to validate the MTT assay data.

Almost all synthesized chalcones were active against the four cancer cell lines, even if some compounds were less potent towards HT-29 CRC cells; thus, chalcones **5**, **9**, **11**, and **15** exhibited IC_50_ values higher than 20 µM at 48 h. Against the three other cell lines, IC_50_ values were in the range of 1–18 µM at 48 h, except chalcones **8** and **10**, which were much more potent, especially towards PCa cells with IC_50_ in the range of 17–31 nM.

The less active compound was 2-pyridone-substituted chalcone **11,** with an IC_50_ superior to 15 µM at 48 h. On the contrary, chalcone **10** exhibited excellent antiproliferative effect as expected; indeed, this chalcone was reported by Kumar et al. as the most potent anticancer agent against human PCa PC3 cells within a series of indolyl chalcones [25]. In addition, its inhibitory activity of cancer cell growth was also shown on breast, lung, ovarian, and hepatic cell lines [26]. Here, we confirmed the great activity of this compound on the PC3 cell line (IC_50_ = 17 nM at 48 h) and demonstrated its potential against another human PCa cell line (IC_50_ = 20 nM on DU145 cells at 48 h) and two human CRC cell lines (IC_50_ = 39 nM and 28 nM on HCT116 and HT-29 cells, respectively, at 48 h) for the first time. Figure 5 showed that indolic chalcone **10** inhibited cell proliferation in a dose-dependent manner. Moreover, the Trypan blue dye exclusion assay may suggest that the effect of this compound is due to a reduced proliferation (Appendix A).

Regarding the series of our new chromonyl chalcones **12**–**15**, we evidenced that this heterocycle at the B ring was able to provide antiproliferative effect (Table 3 and Table 4 and Figure 7). Surprisingly, the cell viability was sometimes higher at 48 h than at 24 h; thus, inhibition of proliferation did not really seem to be time-dependent. This may suggest a mechanism of action different from that of previous chalcones. In addition, the Trypan blue assay may suggest that compound **13** owns an additional impact by inducing toxicity, as shown in Appendix A, for HCT116 cells.

Considering the series of phenolic chalcones, chalcone **8** demonstrated a great antiproliferative activity with IC_50_ in the range of 20 nM against PCa cells. It appeared less active than indolyl chalcone **10** against CRC cells, especially towards HT-29 cells.

Our previous study demonstrated that treatment of HT-29 CRC cells (COX-2 sufficient) with chalcone **8** induced overexpression of COX-2 correlated with an overproduction of PGE2 and enhanced the phosphorylation of p38 MAPK expression, one of the main regulators of COX-2. Indeed, COX-2 overexpression in this case contributed to apoptosis resistance, which may explain the difference in sensitivity to chalcone **8** treatment between HT-29 and HCT116 (COX-2 deficient) CRC cells [2]. In this series, chalcone **2** also revealed interesting results: aside from chalcone **8**, this compound was the most active against HCT116 cells; moreover, it was as potent as chalcone **8** towards HT-29 cells (IC_50_ of 7.3 µM vs. 8.1 µM at 48 h); finally, the IC_50_ values were around 3.0 µM against PCa cells. Both these chalcones present a 3-hydroxy group, which seems to play a positive role in the antiproliferative effect. On the contrary, chalcone **3** bearing a 4-hydroxy group is the less-active phenolic compound against most cell lines.

Therefore, a mechanistic study was carried out to further understand the activities of chalcones **2**, **3**, and **8**. Their ability to impact microtubule depolymerization has been investigated, with chalcone **8** being a reference compound since it is known to be a potent antimitotic agent [18].

#### 2.2.2. Effect of Chalcones on Cellular Microtubules

The effect of chalcones **2**, **3**, and **8** on cellular interphase microtubules was studied using a cell-based assay that quantifies intact microtubules in HeLa cells by immunoluminescence [27].

Our results evidenced that chalcones **2** and **8** induced a dose-dependent depolymerization of cellular microtubules, having, respectively, IC_50_ of 106.2 ± 4.7 µM and 0.289 ± 0.045 µM, whereas compound **3** revealed no effect (Figure 8). These outcomes showed that the most active phenolic chalcones exert their effect by impacting microtubule dynamics: a 3-hydroxy group on the chalcone core was crucial for this interaction while a 4-hydroxy substituent was critical since compound **3** was inactive. Furthermore, by comparing activities of chalcones **2** and **8**, we found that a 4-methoxy group was essential since its removal led to a marked decrease in the effect.

These results were checked using tubulin immunofluorescence on cells to control that the luminescent values do reflect the state of cell microtubules. Two conditions are shown in Figure 9: the immunofluorescence immediately after fixation (−OPT) and the immunofluorescence after a step of free tubulin dimers elimination (+OPT) using warm OPT buffer before fixation, as described in the Materials and Methods Section. As shown in Figure 9, tubulin is mainly in its depolymerized form (−OPT) and only few microtubules remain (+OPT), when cells are treated with chalcone **8** at 1 µM or chalcone **2** at 200 µM, compared with control. These two compounds are therefore confirmed as microtubule-depolymerizing agents.

#### 2.2.3. Effect of Chalcones on Apoptosis-Related Proteins Expression

Considering the fact that chalcones could also exert their antiproliferative effect through pro-apoptotic properties, some inquiries were carried out to investigate the effect of chalcones **10** and **13**, on which the Trypan blue test was performed on apoptosis-related proteins expression. Here, we were interested in the activation of caspase-3 as well as one of its substrates, the poly-ADP-ribose polymerase (PARP). Caspase-3 is described as the main downstream effector caspase that plays an important role in the execution of apoptotic cell death. When caspase-3 is cleaved and thus activated, this enzyme can in turn act on many targets involved in various pathways essential for the cell survival, such as PARP, an enzyme engaged in DNA repair. The cleavage of PARP is considered a hallmark of cells undergoing apoptosis. We compared the expression of procaspase-3, cleaved caspase-3, and native and cleaved PARP forms in treated and untreated cells using Western blotting. As shown in Figure 10 and Figure 11, pro-caspase-3 was highly expressed in the control, but its expression level decreased following the **10** and **13** treatments for all cell lines; this decrease was less marked in DU145 cells. Generated through proteolytic cleavage from pro-caspase-3, the active fragments of caspase-3, namely cleaved caspase-3, were clearly observed in the HCT116, DU145, and PC3 cell lines for both compounds, and in the HT-29 cells for compound **13** only. For compound **10**, cleaved caspase-3 was more weakly produced in HT-29-treated cells compared to the control, in which its expression was mostly undetectable (Figure 10B). Furthermore, results showed that chalcone **10** induced PARP cleavage, as shown by the highly apparent 89 kDa cleaved fragments for the whole cell lines; the same was clearly observed for chalcone **13,** with the exception of the HCT116 cell line. This outcome was associated with a strong decreased expression of the native PARP in treated cells compared to the control for both chalcones. Therefore, these results highlighted apoptosis induction by compounds **10** and **13**.

### 2.3. Structure–Activity Relationship Study

The synthesis of a series of nine phenolic chalcones allowed us to perform a structure–activity relationship study; chalcone **8,** which has already been described [2,4], is the subject of a new evaluation to be compared to the results of the current phenolic compounds (Table 1 and Table 2).

First, the activity of the phenolic chalcones **1**–**3** was studied. On the one hand, it appeared that chalcone **2** was the most active against the whole cell lines, with chalcone **1** being a little less potent. On the other hand, the 4-hydroxy group was found to be critical for the antiproliferative effect, considering the activity of chalcone **3**.

Then, we investigated the influence of an additional methoxy group on biological activity. Considering chalcones with a 3-hydroxy group, we noticed that an additional *para*-methoxy substituent led to a reduction in the activity, while an additional 4-methoxy group was strongly in favor of increasing the effect (Figure 4). These results reinforced the interest of the substitution pattern of combretastatin-A4, i.e., the 3-hydroxy group associated with the 4-methoxy one. In addition, the comparison of IC_50_ values between chalcones **8** and **9** demonstrated that the inversion of these substituents is unfavorable to the effect. This result confirmed the negative impact of a 4-hydroxy group on the antiproliferative activity. Nevertheless, comparing the effects of chalcone **3** relative to those of chalcones **4** and **9**, we evidenced a positive influence of an additional *ortho* or *meta*-methoxy group, except for HT-29 cells (Figure 6).

Now, concerning chalcones with a 2-hydroxy group, conclusions differ depending on the cell lines: in the case of prostatic cells, an additional *para*-methoxy substituent led to a slight increase in the effect as well as an additional 4-OMe group (chalcone **1** vs. chalcones **6** and **7**, Table 2, Figure 3). For the CRC cells (Table 1, Figure 3), this was not the case.

Regarding the series of our new chromonyl chalcones, we evidenced that an additional methyl group at position 6 greatly enhanced the inhibitory activity on cell growth; thus, chromonyl chalcone **13** was four times more active than chalcone **12**, with IC_50_ values in the range of 2.6–5.1 µM. This chalcone was responsible for a cell viability less than 10% at 48 h from the concentration of 10 µM for the whole cell lines (Figure 7). On the contrary, an isopropyl moiety did not improve the antiproliferative effect, since the activity of chalcone **14** was similar to that of chalcone **12**. The results for chalcone **15** were more difficult to interpret. First, we evidenced that this compound was not active against the HT-29 cell line. Then, chalcone **15** demonstrated interesting IC_50_ values against the other cell lines but, surprisingly, as mentioned above, the cell viability was sometimes higher at 48 h than it was at 24 h (Figure 7).

## 3. Materials and Methods

### 3.1. Chemical Synthesis

#### 3.1.1. General Chemistry

Unless otherwise specified, all chemicals were purchased from Fischer Scientific (Illkirch, France), Sigma-Aldrich (St Quentin Fallavier, France) and TCI (Paris, France), and used without further purification.

All reactions were monitored by TLC on silica plates (0.20 mm silica gel 60 with UV254 indicator, Sigma-Aldrich) and visualized using a combination of UV light. Purifications were performed using an automated purification system Interchim Puriflash 430, using 200–800 nm UV scan as a detector (Interchim, Montluçon, France), and a mixture of cyclohexane/ethyl acetate (EtOAc) as eluent. NMR spectra were recorded on a Bruker Advance III 300 MHz spectrometer (Bruker, Wissembourg, France) operating at 300 MHz for ^1^H and 75 MHz for ^13^C. Chemical shifts were expressed in δ (ppm) values relative to tetramethylsilane (TMS) used as an internal reference, and coupling constants (*J*) were expressed in Hz. High-resolution mass spectra (HRMS) were measured on an Agilent 6530 QTOF-LC/MS mass analyzer (Agilent, Les Ulis, France) in ESI+. Melting points were recorded on a scientific analyzer SMP 10 apparatus (Cole-Parmer, Villepinte, France) and were uncorrected. FT-IR spectra were recorded with a PerkinElmer Frontier (PerkinElmer, Villebon-sur-Yvette, France.

#### 3.1.2. General Procedure for the Synthesis of Chalcones **1**–**11** in Basic Conditions

Compounds **1**–**11** were synthesized by a Claisen–Schmidt condensation using ethanol as solvent and a base (NaOH or piperidine). 3′,4′,5′-trimethoxyacetophenone (1 equiv.) was dissolved in ethanol in a round-bottomed flask with the base. After 30 min of stirring at room temperature, the corresponding substituted aldehyde (1.2 equiv.) was added dropwise, and the reaction mixture was run at room temperature or reflux for a varying duration under continued stirring. Then, the reaction mixture was neutralized with aqueous hydrochloric acid (1 M). When the product precipitated (conditions A), the latter was filtered and washed with cold ethanol and cold distilled water, leading to pure chalcones. The filtrates were collected and purified by recrystallization in ethanol. When there was no precipitation (conditions B), the aqueous phase was thrice extracted with EtOAc. The organic phase was washed with a saturated aqueous solution of sodium chloride and dried on MgSO_4_; after filtration, the solvent was evaporated under a vacuum. The residue was purified by flash chromatography in a mixture of cyclohexane/EtOAc.

The general structure of chalcones **1**–**9** is as follows:



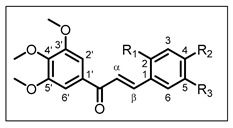



(*E*)-3-(2-hydroxyphenyl)-1-(3,4,5-trimethoxyphenyl)prop-2-en-1-one **1** (R_1_ = OH, R_2_ = R_3_ = H): NaOH (100 mmol, 25 equiv.) was used as the base and the reaction, using 2-hydroxybenzaldehyde (4.8 mmol), was refluxed for 4 h (conditions B). Yield 47%; yellow solid; R_f_ = 0.37, eluent (Cyclohexane/EtOAc 6/4); ^1^H NMR (300 MHz, CDCl_3_, 25 °C) δ, 8.21 (1H, d, *J* = 15.8 Hz, H-β), 7.65 (1H, d, *J* = 15.8 Hz, H-α), 7.58 (1H, br d, *J* = 7.8 Hz, H-6), 7.21–7.31 (3H, m, H-2′, H-6′ and H-4), 6.88–7.00 (2H, m, H-3 and H-5), 3.93 (3H, s, 4′-OMe), 3.92 (6H, s, 3′-OMe and 5′-OMe). ^13^C NMR (75 MHz, CDCl_3_, 25 °C) δ, 190.9 (CO), 156.1 (C-2), 153.2 (C-3′ and C-5′), 142.5 (C-4′), 141.1 (C-β), 133.7 (C-1′), 132.0 (C-4), 129.6 (C-6), 122.7 (C-α), 122.3 (C-1), 120.9 (C-3), 116.8 (C-5), 106.4 (C-2′ and C-6′), 61.1 (4′-OMe), 56.5 (3′-OMe and 5′-OMe). HRMS (ESI+) [M + Na]^+^ calcd. for C_18_H_18_O_5_Na: *m/z* 337.1052, found *m/z* 337.1083.

(*E*)-3-(3-hydroxyphenyl)-1-(3,4,5-trimethoxyphenyl)prop-2-en-1-one 2 (R_1_ = R_2_ = H, R_3_ = OH): NaOH (10 mmol, 2.5 equiv.) was used as the base and the reaction, using 3-hydroxybenzaldehyde (4.8 mmol), was refluxed for 1.5 h (conditions B). Yield 80%; yellow solid; R_f_ = 0.40, eluent (Cyclohexane/EtOAc 6/4); ^1^H NMR (300 MHz, CDCl_3_, 25 °C) δ, 7.77 (1H, d, *J* = 15.6 Hz, H-β), 7.45 (1H, d, *J* = 15.6 Hz, H-α), 7.25–7.32 (3H, m, H-2′, H-6′ and H-5), 7.15–7.23 (2H, m, H-2 and H-6), 6.95 (1H, br d, *J* = 8.0 Hz, H-4), 3.94 (3H, s, 4′-OMe), 3.93 (6H, s, 3′-OMe and 5′-OMe). ^13^C NMR (75 MHz, CDCl_3_, 25 °C) δ, 189.9 (CO), 156.5 (C-3), 153.3 (C-3′ and C-5′), 145.1 (C-β), 142.7 (C-4′), 136.4 (C-1), 133.4 (C-1′), 130.3 (C-5), 122.1 (C-α), 121.1 (C-6), 118.1 (C-4), 115.2 (C-2), 106.3 (C-2′ and C-6′), 61.2 (4′-OMe), 56.5 (3′-OMe and 5′-OMe). HRMS (ESI+) [M + H]^+^ calcd. for C_18_H_19_O_5_: *m*/*z* 315.1233, found *m*/*z* 315.1247.

(*E*)-3-(4-hydroxyphenyl)-1-(3,4,5-trimethoxyphenyl)prop-2-en-1-one 3 (R_1_ = R_3_ = H, R_2_ = OH): NaOH (25 mmol, 25 equiv.) was used as the base and the reaction, using 4-hydroxybenzaldehyde (1.2 mmol), was refluxed for 5 h (conditions B). Yield 32%; yellow solid; R_f_ = 0.32, eluent (Cyclohexane/EtOAc 6/4); ^1^H NMR (300 MHz, CDCl_3_, 25 °C) δ, 7.79 (1H, d, *J* = 15.6 Hz, H-β), 7.58 (2H, d, *J* = 8.5 Hz, H-2 and H-6), 7.35 (1H, d, *J* = 15.6 Hz, H-α), 7.27 (2H, s, H-2′ and H-6′), 6.93 (2H, d, *J* = 8.5 Hz, H-3 and H-5), 3.94 (9H, s, 4′-OMe, 3′-OMe and 5′-OMe). ^13^C NMR (75 MHz, CDCl_3_, 25 °C) δ, 189.9 (CO), 158.5 (C-4), 153.3 (C-3′ and C-5′), 145.2 (C-β), 142.5 (C-4′), 133.9 (C-1′), 130.7 (C-2 and C-6), 127.6 (C-1), 119.4 (C-α), 116.2 (C-3 and C-5), 106.2 (C-2′ and C-6′), 61.2 (4′-OMe), 56.5 (3′-OMe and 5′-OMe). HRMS (ESI+) [M + H]^+^ calcd. for C_18_H_19_O_5_: *m*/*z* 315.1233, found *m*/*z* 315.1297.

(*E*)-3-(4-hydroxy-2-methoxyphenyl)-1-(3,4,5-trimethoxyphenyl)prop-2-en-1-one 4 (R_1_ = OMe, R_2_ = OH, R_3_ = H): NaOH (10 mmol, 2.5 equiv.) was used as the base and the reaction, using 4-hydroxy-2-methoxybenzaldehyde (4.8 mmol), was run at room temperature for 6 days (conditions B). Yield 19%; yellow solid; R_f_ = 0.24, eluent (Cyclohexane/EtOAc 6/4); m.p. 168–170 °C; ^1^H NMR (300 MHz, CDCl_3_, 25 °C) δ, 8.06 (1H, d, *J* = 15.8 Hz, H-β), 7.50 (1H, d, *J* = 8.3 Hz, H-6), 7.45 (1H, d, *J* = 15.8 Hz, H-α), 7.27 (2H, s, H-2′ and H-6′), 6.44–6.68 (2H, m, H-3 and H-5), 3.93 (9H, s, 3′-OMe, 4′-OMe and 5′-OMe), 3.84 (3H, s, 2-OMe). ^13^C NMR (75 MHz, CDCl_3_, 25 °C) δ, 190.9 (CO), 160.8 (C-2), 160.2 (C-4), 153.2 (C-3′ and C-5′), 142.3 (C-4′), 141.2 (C-β), 134.2 (C-1′), 131.1 (C-6), 120.0 (C-α), 116.8 (C-1), 108.3 (C-5), 106.6 (C-2′ and C-6′), 99.4 (C-3), 61.1 (4′-OMe), 56.5 (3′-OMe and 5′-OMe), 55.7 (2-OMe). HRMS (ESI+) [M + H]^+^ calcd. for C_19_H_21_O_6_: *m*/*z* 345.1338, found *m*/*z* 345.1346; IR (ν_max_/cm^−1^) 3358, 1642, 1547, 1504, 1280, 1157, 1120, 1000, 828.

(*E*)-3-(5-hydroxy-2-methoxyphenyl)-1-(3,4,5-trimethoxyphenyl)prop-2-en-1-one 5 (R_1_ = OMe, R_2_ = H, R_3_ = OH): NaOH (1.25 mmol, 2.5 equiv.) was used as the base and the reaction, using 5-hydroxy-2-methoxybenzaldehyde (0.6 mmol), was run at room temperature for 2 days (conditions B). Yield 50%; yellow solid; R_f_ = 0.36, eluent (Cyclohexane/EtOAc 6/4); m.p. 136–138 °C; ^1^H NMR (300 MHz, CDCl_3_, 25 °C) δ, 8.04 (1H, d, *J* = 15.8 Hz, H-β), 7.48 (1H, d, *J* = 15.8 Hz, H-α), 7.24 (2H, s, H-2′ and H-6′), 7.15 (1H, d, *J* = 2.9 Hz, H-6), 6.89 (1H, dd, *J* = 8.9 Hz and *J* = 2.9 Hz, H-4), 6.80 (1H, d, *J* = 8.9 Hz, H-3), 3.92 (3H, s, 4′-OMe), 3.91 (6H, s, 3′-OMe and 5′OMe), 3.83 (3H, s, 2-OMe). ^13^C NMR (75 MHz, CDCl_3_, 25 °C) δ (ppm), 190.4 (CO), 153.3 (C-2), 153.2 (C-3′ and C-5′), 149.8 (C-5), 142.5 (C-4′), 140.4 (C-β), 133.7 (C-1′), 124.7 (C-1), 123.1 (C-α), 118.8 (C-4), 115.4 (C-6), 112.8 (C-3), 106.3 (C-2′ and C-6′), 61.1 (4′-OMe), 56.5 (3′-OMe and 5′-OMe), 56.3 (2-OMe). HRMS (ESI+) [M + H]^+^ calcd. for C_19_H_21_O_6_: *m*/*z* 345.1338, found *m*/*z* 345.1402; IR (ν_max_/cm^−1^) 3284, 1642, 1563, 1494, 1299, 1158, 1123, 1002, 971, 801.

(*E*)-3-(2-hydroxy-5-methoxyphenyl)-1-(3,4,5-trimethoxyphenyl)prop-2-en-1-one 6 (R_1_ = OH, R_2_ = H, R_3_ = OMe): NaOH (10 mmol; 2.5 equiv.) was used as the base and the reaction, using 2-hydroxy-5-methoxybenzaldehyde (4.8 mmol), was refluxed for 15 h (conditions B). Yield 10%; yellow solid; R_f_ = 0.38, eluent (Cyclohexane/EtOAc 6/4); ^1^H NMR (500 MHz, CDCl_3_, 25 °C) δ, 8.07 (1H, d, *J* = 15.7 Hz, H-β), 7.57 (1H, d, *J* = 15.7 Hz, H-α), 7.28 (2H, s, H-2′ and H-6′), 7.10 (1H, d, *J* =2.9 Hz, H-6), 7.57 (1H, d, *J* = 15.7 Hz, H-α), 6.84 (1H, dd, *J* = 8.8 Hz and *J* = 2.9 Hz, H-4), 6.83 (1H, d, *J* = 8.8 Hz, H-3), 3.94 (9H, s, 3′-OMe, 4′-OMe and 5′-OMe), 3.81 (3H, s, 5-OMe). ^13^C NMR (125 MHz, CDCl_3_, 25 °C) δ, 190.5 (CO), 153.8 (C-5), 153.3 (C-3′ and C-5′), 149.9 (C-2), 142.6 (C-4′), 140.5 (C-β), 133.7 (C-1′), 123.2 (C-α), 122.9 (C-1), 117.6 (C-4), 113.7 (C-6), 106.5 (C-2′ and C-6′), 61.1 (4′-OMe), 56.6 (3′-OMe and 5′-OMe), 56.1 (5-OMe). HRMS (ESI+) [M + H]^+^ calcd. for C_19_H_21_O_6_: *m*/*z* 345.1338, found *m*/*z* 345.1329.

(*E*)-3-(2-hydroxy-4-methoxyphenyl)-1-(3,4,5-trimethoxyphenyl)prop-2-en-1-one 7 (R_1_ = OH, R_2_ = OMe, R_3_ = H): NaOH (25 mmol, 25 equiv.) was used as the base and the reaction, using 2-hydroxy-4-methoxybenzaldehyde (1.2 mmol), was refluxed for 20 h (conditions B). Yield 10%; yellow solid; R_f_ = 0.18, eluent (Cyclohexane/EtOAc 6/4); ^1^H NMR (500 MHz, CDCl_3_, 25 °C) δ, 8.09 (1H, d, *J* = 15.7 Hz, H-β), 7.52 (1H, d, *J* = 8.6 Hz, H-6), 7.51 (1H, d, *J* = 15.7 Hz, H-α), 7.27 (2H, s, H-2′ and H-6′), 6.52 (1H, dd, *J* = 8.6 Hz and *J* = 2.3 Hz, H-5), 6.48 (1H, d, *J* = 2.3 Hz, H-3), 3.93 (9H, s, 3′-OMe, 4′-OMe and 5′-OMe), 3.81 (3H, s, 4-OMe). ^13^C NMR (125 MHz, CDCl_3_, 25 °C) δ, 190.7 (CO), 163.1 (C-4), 157.7 (C-2), 153.2 (C-3′ and C-5′), 142.4 (C-4′), 141.0 (C-β), 134.1 (C-1′), 131.0 (C-6), 120.1 (C-α), 115.5 (C-1), 107.6 (C-5), 106.4 (C-2′ and C-6′), 102.2 (C-3), 61.1 (4′-OMe), 56.5 (3′-OMe and 5′-OMe), 55.6 (4-OMe). HRMS (ESI+) [M + H]^+^ calcd. for C_19_H_21_O_6_: *m*/*z* 345.1338, found *m*/*z* 345.1331.

(*E*)-3-(3-hydroxy-4-methoxyphenyl)-1-(3,4,5-trimethoxyphenyl)prop-2-en-1-one 8 (R_1_ = H, R_2_ = OMe, R_3_ = OH): NaOH (5 mmol, 5 equiv.) was used as the base and the reaction, using 3-hydroxy-4-methoxybenzaldehyde (1.2 mmol), was refluxed for 2 h (conditions B). Yield 69%; yellow solid; R_f_ = 0.38, eluent (Cyclohexane/EtOAc 6/4); ^1^H NMR (300 MHz, CDCl_3_, 25 °C) δ, 7.74 (1H, d, *J* = 15.5 Hz, H-β), 7.35 (1H, d, *J* = 15.5 Hz, H-α), 7.30 (1H, d, *J* = 2.1 Hz, H-2), 7.21–7.27 (2H, s, H-2′ and H-6′), 7.13 (1H, dd, *J* = 8.3 Hz and *J* = 2.1 Hz, H-6), 6.88 (1H, d, *J* = 8.3 Hz, H-5), 3.95 (6H, s, 3′-OMe and 5′-OMe), 3.94 (3H, s, 4-OMe), 3.93 (3H, s, 4′-OMe). ^13^C NMR (75 MHz, CDCl_3_, 25 °C) δ, 189.3 (CO), 152.2 (C-3′ and C-5′), 149.0 (C-4), 146.1 (C-3), 144.8 (C-β), 142.5 (C-4′), 133.9 (C-1′), 128.7 (C-1), 123.1 (C-6), 120.1 (C-α), 112.9 (C-2), 110.8 (C-5), 106.2 (C-2′ and C-6′), 61.1 (4′-OMe), 56.5 (3′-OMe and 5′-OMe), 56.2 (4-OMe).

(*E*)-3-(4-hydroxy-3-methoxyphenyl)-1-(3,4,5-trimethoxyphenyl)prop-2-en-1-one 9 (R_1_ = H, R_2_ = OH, R_3_ = OMe): NaOH (25 mmol, 25 equiv.) was used as the base and the reaction, using 4-hydroxy-3-methoxybenzaldehyde (1.2 mmol), was refluxed for 5 h (conditions B). Yield 12%; yellow solid; R_f_ = 0.35, eluent (Cyclohexane/EtOAc 6/4); ^1^H NMR (500 MHz, CDCl_3_, 25 °C) δ, 7.73 (1H, d, *J* = 15.6 Hz, H-β), 7.30 (1H, d, *J* = 15.6 Hz, H-α), 7.24 (2H, s, H-2′ and H-6′), 7.22- (1H, dd, *J* = 8.2 Hz and *J* = 1.9 Hz, H-6), 7.10 (1H, d, *J* = 1.9 Hz, H-2), 6.95 (1H, d, *J* = 8.2 Hz, H-5), 5.94 (1H, br s, OH), 3.94 (3H, s, 3-OMe), 3.93 (6H, s, 3′-OMe and 5′-OMe), 3.92 (3H, s, 4′-OMe). ^13^C NMR (125 MHz, CDCl_3_, 25 °C) δ, 189.7 (CO), 153.4 (C-3′ and C-5′), 148.5 (C-3), 147.0 (C-4), 145.3 (C-β), 142.6 (C-4′), 134.1 (C-1′), 127.7 (C-1), 123.2 (C-6), 119.8 (C-α), 115.1 (C-5), 110.7 (C-2), 106.4 (C-2′ and C-6′), 61.2 (4′-OMe), 56.7 (3′-OMe and 5′-OMe), 56.3 (3-OMe).

The structure of chalcone **10** is as follows:



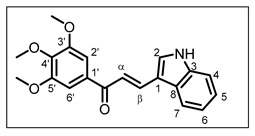



(*E*)-3-(1*H*-indol-3-yl)-1-(3,4,5-trimethoxyphenyl)prop-2-en-1-one **10** [25]: Piperidine (20 mmol, 5 equiv.) was used as the base and the reaction, using 1*H*-indole-3-carbaldehyde (4.8 mmol), was refluxed for 3 days. Before the neutralization, the mixture was poured into crushed ice (conditions A). Yield 79%; yellow solid; R_f_ = 0.73, eluent (DCM/EtOAc 6/4); ^1^H NMR (300 MHz, DMSO-*d_6_*, 25 °C) δ, 11.93 (1H, br s, NH), 8.16 (1H, br s, H-2), 8.00–8.10 (2H, m, H-7 and H-β), 7.63 (1H, d, *J* = 15.4 Hz, H-α), 7.46–7.52 (1H, m, H-4), 7.37 (2H, s, H-2′ and H-6′), 7.18–7.28 (2H, m, H-5 and H-6), 3.91 (6H, s, H-3′-OMe and H-5′-OMe), 3.76 (3H, s, H-4′-OMe). ^13^C NMR (75 MHz, DMSO-*d_6_*, 25 °C) δ, 188.4 (CO), 153.3 (C-3′ and C-5′), 141.8 (C-4′), 139.1 (C-β), 137.9 (C-3), 134.5 (C-1′), 133.2 (C-2), 125.8 (C-8), 123.1 (C-5), 121.6 (C-6), 120.7 (C-7), 116.0 (C-α), 113.2 (C-1), 112.9 (C-4), 106.2 (C-2′ and C-6′), 60.6 (C-4′-OMe), 56.6 (C-3′-OMe and C-5′-OMe). HRMS (ESI+) [M + H]^+^ calcd. for C_20_H_20_NO_4_: *m***/***z* 338.1392, found *m***/***z* 338.1458; IR (ν_max_/cm^−1^) 3245, 1644, 1552, 1351, 1246, 1129, 811, 721, 699.

The structure of chalcone **11** is as follows:



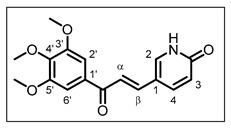



(*E*)-5-(3-oxo-3-(3,4,5-trimethoxyphenyl)prop-1-en-1-yl)pyridin-2(1*H*)-one **11**: NaOH (10 mmol, 2.5 equiv.) was used as the base and the reaction, using 6-oxo-1,6-dihydropyridine-3-carbaldehyde (4.8 mmol), was run at room temperature for 6 days (conditions B). Yield 18%; yellow solid; R_f_ = 0.10, eluent (Cyclohexane/EtOAc 3/7); m.p. 217–219 °C; ^1^H NMR (300 MHz, DMSO-*d_6_*, 25 °C) δ, 12.06 (1H, br s, NH), 8.20 (1H, dd, *J* = 9.7 Hz and *J* = 2.6 Hz, H-4), 7.98 (1H, d, *J* = 2.6 Hz, H-2), 7.66 (1H, d, *J* = 15.6 Hz, H-α), 7.61 (1H, d, *J* = 15.6 Hz, H-β), 7.39 (2H, s, H-2′ and H-6′), 6.45 (1H, d, *J* = 9.7 Hz, H-3), 3.89 (6H, s, 3′-OMe and 5′-OMe), 3.75 (3H, s, 4′-OMe). ^13^C NMR (75 MHz, DMSO-*d_6_*, 25 °C) δ (ppm), 187.4 (CO), 162.2 (CONH), 152.9 (C-3′ and C-5′), 141.7 (C-4′), 140.6 (C-β), 140.2 (C-2), 138.4 (C-4), 133.3 (C-1′), 120.3 (C-3), 117.8 (C-α), 114.0 (C-1), 106.0 (C-2′ and C-6′), 60.2 (4′-OMe), 56.2 (3′-OMe and 5′-OMe). HRMS (ESI+) [M + Na]^+^ calcd. for C_17_H_17_NO_5_Na: *m***/***z* 338.1004, found *m***/***z* 338.1045; IR (ν_max_/cm^−1^) 2833, 1656, 1572, 1160, 1111, 972, 832.

#### 3.1.3. General Procedure for the Synthesis of Chalcones **3** and **9** Using a Protection Step

The corresponding benzaldehyde (1 equiv.) was dissolved in CH_2_Cl_2_ with pyridinium *p*-toluenesulfonate (PPTS, 0.025 equiv.). After 30 min of stirring at room temperature, 3,4-dihydro-2*H*-pyrane (DHP, 3 equiv.) was added and the reaction was run for 24 h at room temperature. Then, the organic phase was washed with alkaline water (NaHCO_3_ 5%) thrice and dried on Na_2_SO_4_; after a Büchner filtration, the solvent was evaporated under a vacuum. The expected product was obtained without purification as a colourless oil in a quantitative yield. The protected benzaldehyde reacted with 3,4,5-trimethoxyacetophenone in basic conditions as previously described. After purification by flash chromatography, the protected chalcone was dissolved in MeOH with p-toluenesulfonic acid (PTSA, 0.1 equiv.). After 3 h of stirring at room temperature, methanol was evaporated under a vacuum; the residue was diluted in 20 mL of H_2_O. The aqueous phase was thrice extracted with 20 mL of CH_2_Cl_2_. The organic phase was dried on MgSO_4_ and, after a Büchner filtration, the solvent was evaporated under a vacuum. The deprotection step afforded the corresponding chalcones in a quantitative yield. The global yield for chalcone **3** was 68%, while for chalcone **9**, it was 37%.

#### 3.1.4. General Procedure for the Synthesis of Chalcones **12**–**15** in Acidic Conditions

Compounds **12**–**15** were synthesized by a Claisen–Schmidt condensation using EtOH as solvent and PTSA 12% in acetic acid as catalyst. 3′,4′,5′-trimethoxyacetophenone (1 equiv.) and the corresponding substituted aldehyde (1.2 equiv.) was dissolved in ethanol in a round-bottomed flask topped by an air-cooling column. A few drops of PTSA 12% in acetic acid were added, and the reaction mixture was refluxed for a varying duration under continued stirring. The reaction mixture was allowed to cool down to room temperature and the product precipitated. The latter was filtered and washed with cold ethanol and cold distilled water, leading to chalcones in a pure form. The filtrates were collected and purified by recrystallization in ethanol.

The general structure of chalcones **12**–**15** is as follows:



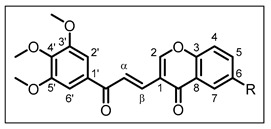



(*E*)-3-(3-oxo-3-(3,4,5-trimethoxyphenyl)prop-1-en-1-yl)-4*H*-chromen-4-one **12** (R = H). The reaction was run for 17 h, using 4-oxo-4*H*-chromene-3-carbaldehyde (0.72 mmol). Yield 60%; pale-yellow solid; R_f_ = 0.52, eluent (Cyclohexane/EtOAc 6/4); m.p. 215–217 °C; ^1^H NMR (300 MHz, CDCl_3_, 25 °C) δ 8.63 (1H, d, *J* = 15.2 Hz, H-α), 8.31 (1H, dd, *J* = 8.0 Hz and *J* = 1.7 Hz, H-7), 8.22 (1H, s, H-2), 7.73 (1H, td, *J* = 8.0 Hz and *J* = 1.7 Hz, H-5), 7.47–7.57 (2H, m, H-4 and H-6), 7.48 (1H, d, *J* = 15.2 Hz, H-β), 7.37 (2H, s, H-2′ and H-6′), 3.97 (6H, s, 3′-OMe and 5′-OMe), 3.94 (3H, s, 4′-OMe). ^13^C NMR (75 MHz, CDCl_3_, 25 °C) δ, 189.6 (CO), 176.5 (CO, chromone), 159.1 (C-2), 155.6 (C-3), 153.3 (C-3′ and C-5′), 142.7 (C-4′), 135.5 (C-β), 134.2 (C-5), 133.3 (C-1′), 126.4 (C-7), 126.1 (C-6), 125.6 (C-α), 124.4 (C-8), 119.7 (C-1), 118.3 (C-4), 106.3 (C-2′ and C-6′), 61.1 (4′-OMe), 56.5 (3′-OMe and 5′-OMe). HRMS (ESI+) [M + Na]^+^ calcd. for C_21_H_18_O_6_Na: *m*/*z* 389.1001, found *m*/*z* 389.1031; IR (ν_max_/cm^−1^) 1657, 1573, 1561, 1461, 1412, 1338, 1309, 1285, 1159, 1125, 993, 847, 770, 727.

(*E*)-6-methyl-3-(3-oxo-3-(3,4,5-trimethoxyphenyl)prop-1-en-1-yl)-4H-chromen-4-one 13 (R = Me). The reaction was run for 8 days, using 6-methyl-4-oxo-4*H*-chromene-3-carbaldehyde (1.2 mmol). Quantitative yield; pale yellow solid; R_f_ = 0.62, eluent (Cyclohexane/EtOAc 6/4); m.p. 176–178 °C; ^1^H NMR (300 MHz, CDCl_3_, 25 °C) δ, 8.63 (1H, d, *J* = 15.2 Hz, H-α), 8.19 (1H, s, H-2), 8.08 (1H, br s, H-7), 7.52 (1H, dd, *J* = 8.8 Hz and *J* = 1.9 Hz, H-5), 7.48 (1H, d, *J* = 15.2 Hz, H-β), 7.40 (1H, d, *J* = 8.8 Hz, H-4), 7.37 (2H, s, H-2′ and H-6′), 3.97 (6H, s, 3′-OMe and 5′-OMe), 3.94 (3H, s, 4′-OMe), 2.48 (3H, s, 6-Me). ^13^C NMR (75 MHz, CDCl_3_, 25 °C) δ, 189.6 (CO), 176.6 (CO, chromone), 159.1 (C-2), 153.9 (C-3), 153.3 (C-3′ and C-5′), 142.7 (C-4′), 136.2 (C-6), 135.7 (C-β), 135.5 (C-5), 133.4 (C-1′), 125.7 (C-7), 125.4 (C-α), 124.1 (C-8), 119.5 (C-1), 118.1 (C-4), 106.3 (C-2′ and C-6′), 61.1 (4′-OMe), 56.5 (3′-OMe and 5′-OMe), 21.2 (6-Me). HRMS (ESI+) [M + H]^+^ calcd. for C_22_H_21_O_6_: *m*/*z* 381.1338, found *m*/*z* 381.1359; IR (ν_max_/cm^−1^) 1661, 1582, 1482, 1339, 1310, 1286, 1125, 1010, 806, 724.

(*E*)-6-isopropyl-3-(3-oxo-3-(3,4,5-trimethoxyphenyl)prop-1-en-1-yl)-4*H*-chromen-4-one 14 (R = iPr). The reaction was run for 3 days, using 6-isopropyl-4-oxo-4*H*-chromene-3-carbaldehyde (1.2 mmol). Yield 70%; pale yellow solid; R_f_ = 0.60, eluent (Cyclohexane/EtOAc 6/4); m.p. 183–185 °C; ^1^H NMR (300 MHz, CDCl_3_, 25 °C) δ, 8.64 (1H, d, *J* = 15.5 Hz, H-α), 8.20 (1H, s, H-2), 8.15 (1H, d, *J* = 2.2 Hz, H-7), 7.60 (1H, dd, *J* = 8.9 Hz and *J* = 2.2 Hz, H-5), 7.49 (1H, d, *J* = 15.5 Hz, H-β), 7.45 (1H, d, *J* = 8.9 Hz, H-4), 7.38 (2H, s, H-2′ and H-6′), 3.97 (6H, s, 3′-OMe and 5′-OMe), 3.94 (3H, s, 4′-OMe), 3.07 (1H, hept, *J* = 6.9 Hz, 6-CH (iPr)), 1.32 (6H, d, *J* = 6.9 Hz, 2 x 6-Me (iPr)). ^13^C NMR (75 MHz, CDCl_3_, 25 °C) δ, 189.6 (CO), 176.8 (CO, chromone), 159.1 (C-2), 154.1 (C-3), 153.3 (C-3′ and C-5′), 147.2 (C-6), 142.7 (C-4′), 135.8 (C-β), 133.4 (C-1′), 133.3 (C-5), 125.4 (C-α), 124.2 (C-8), 123.1 (C-7), 119.5 (C-1), 118.2 (C-4), 106.3 (C-2′ and C-6′), 61.1 (4′-OMe), 56.5 (3′-OMe and 5′-OMe), 34.0 (6-CH (iPr)), 24.1 (2x 6-Me (iPr)). HRMS (ESI+) [M + H]^+^ calcd. for C_24_ H_25_O_6_: *m*/*z* 409.1651, found *m*/*z* 409.1692; IR (ν_max_/cm^−1^) 1657, 1578, 1564, 1480, 1338, 1311, 1279, 1127, 1003, 824, 728.

(*E*)-6-hydroxy-3-(3-oxo-3-(3,4,5-trimethoxyphenyl)prop-1-en-1-yl)-4*H*-chromen-4-one 15 (R = OH). The reaction was run for 10 days, using 6-hydroxy-4-oxo-4*H*-chromene-3-carbaldehyde (1.2 mmol). Yield 86%; pale yellow solid; R_f_ = 0.10, eluent (Cyclohexane/EtOAc 6/4); m.p. 294–296 °C; ^1^H NMR (300 MHz, DMSO-*d_6_*, 25 °C) δ, 10.18 (1H, s, OH), 9.02 (1H, s, H-2), 8.32 (1H, d, *J* = 15.6 Hz, H-α), 7.63 (1H, d, *J* = 15.6 Hz, H-β), 7.61 (1H, d, *J* = 9.0 Hz, H-4), 7.42 (1H, d, *J* = 3.0 Hz, H-7), 7.32 (2H, s, H-2′ and H-6′), 7.28 (1H, dd, *J* = 9.0 Hz and *J* = 3.0 Hz, H-5), 3.90 (6H, s, 3′-OMe and 5′-OMe), 3.77 (3H, s, 4′-OMe). ^13^C NMR (75 MHz, DMSO-*d_6_*, 25 °C) δ, 188.3 (CO), 175.2 (CO, chromone), 159.9 (C-2), 155.4 (C-6), 152.9 (C-3′ and C-5′), 149.0 (C-3), 142.0 (C-4′), 136.0 (C-β), 132.9 (C-1′), 125.4 (C-5), 124.4 (C-8), 122.9 (C-α), 120.0 (C-4), 108.1 (C-7), 117.6 (C-1), 105.9 (C-2′ and C-6′), 60.2 (4′-OMe), 56.2 (3′-OMe and 5′-OMe). HRMS (ESI+) [M + H]^+^ calcd. for C_21_H_19_O_7_: *m*/*z* 383.1130, found *m*/*z* 383.1122; IR (ν_max_/cm^−1^) 3294, 1657, 1576, 1470, 1336, 1284, 1228, 1126, 1005, 986, 843, 733.

### 3.2. In Vitro Biological Assays

#### 3.2.1. Materials

RPMI 1640 medium, fetal bovine serum (FBS), L-glutamine, penicillin-streptomycin, and phosphate-buffered saline (PBS) were purchased from Gibco BRL—Fisher Scientific (Illkirch, France). 3-(4,5-dimethylthiazol-2-yl)-2,5-diphenyltetrazolium bromide (MTT), DMSO, Trypan blue, Immobilon Western Chemiluminescent HRP Substrate, and human anti-β-actin antibody were obtained from Sigma-Aldrich—Merck (Saint-Quentin-Fallavier, France). Pro-caspase-3, cleaved caspase-3, poly-ADP-ribose polymerase (PARP) antibodies, and goat anti-rabbit secondary antibody conjugated to horse-radish peroxidase (HRP) were purchased from Cell Signaling Technology—Ozyme (Saint-Quentin-en-Yvelines, France). Rabbit anti-mouse conjugated to HRP secondary antibody were obtained from Invitrogen—Thermo Fisher Scientific (Villebon-Sur-Yvette, France).

#### 3.2.2. Cell Lines and Culture Conditions

Human CRC (HCT116 and HT-29) and PCa (DU145 and PC3) cell lines were purchased from the American Type Culture Collection (ATCC—LGC Standards, Molsheim, France). All cells were grown in RPMI 1640 medium supplemented with 10% FBS, 1% L-glutamine, 100 U/mL penicillin, and 100 μg/mL streptomycin. Cultures were maintained in a humidified atmosphere containing 5% CO_2_ at 37 °C. Cells were seeded in 75 cm^2^ culture flasks at 1.2 × 10^4^ cells/cm^2^ for HCT116, DU145, and PC3 cells, and 2.0 × 10^4^ for HT-29 cells. Cells were grown for 24 h in culture medium prior to exposure or not to obtain the synthesized compounds. Stock solutions (10^−2^ M in DMSO) of each compound were diluted in culture medium to obtain appropriate final concentrations. The same amount of the vehicle (percentage of DMSO did not exceed 0.5%) was added to the control cells.

#### 3.2.3. Cell Viability

(a)MTT assay: The antiproliferative effect of all compounds was determined using MTT assay (a metabolic activity test). Briefly, cells were seeded in 96-well culture plates at 4 × 10^3^ cells/well for HCT116, PC3, and DU145 cells and 6 × 10^3^ cells/well for HT-29 cells (100 µL/well), and grown for 24 h in culture medium prior to exposure or not to obtain compounds **1**–**15** with concentrations ranging from 5 nM to 50 μM. After 24 and 48 h of treatment, MTT (5 mg/mL in PBS) was added and incubated for another 3 h. MTT was then removed from the wells before adding 100 μL/well of DMSO. The optical density was detected with a microplate reader (Multiskan FC, Thermoscientific, Bordeaux, France) at 550 nm and cell viability was expressed as a percentage of each treatment condition compared to control cells. IC_50_ values were calculated for all compounds from the dose–response curve. Data are expressed as the arithmetic means ± standard error of the mean (SEM) of separate experiments. All experiments were performed at least in triplicate.(b)Trypan blue dye exclusion method: Cells were grown for 24 h then treated with compounds **10** and **13** at indicated concentrations. After 48 h of treatment, cells were trypsinized and resuspended in complete medium. Each sample was mixed with Trypan blue solution (0.14% in HBSS, Invitrogen-GIBCO, ThermoFisher, Illkirch, France). Colored (non-viable) and dye-excluding (viable) cells were counted with the LUNA-II automated cell Counter (Logos Biosystems; MC2, Aubière, France). The control was normalized to 100% including viable and dead cells. The results indicated on the histograms were expressed as the percentage of cells compared to the control. IC_50_ values were determined from the percentages of viable cells compared to the control.

#### 3.2.4. Quantitative Assay of the Cellular Microtubule Content

The assay was realized as described by Laisne et al. [27]. Briefly, 7500 HeLa cells were seeded in 96-well microplates (#655086, Greiner bioOne, Courtaboeuf, France) in 100 µL of complete medium per well and then incubated at 37 °C in 5% CO_2_ for 24 h. Cells were then treated for 30 min at 37 °C with the compounds at concentrations ranging from 0.1 to 1000 µM (1 microplate per molecule, 1 concentration per column), with 0.1% DMSO used as positive control (6 wells per microplate). After medium aspiration, treated cells were permeabilized for 10 min using 100 µL per well of warmed (37 °C) OPT buffer (80 mM Pipes, 1 mM EGTA, 1 mM MgCl_2_, 0.5% Triton X-100, and 10% glycerol, pH 6.8). Cells were fixed overnight at room temperature using 100 µL per well of 4% formaldehyde (Sigma Aldrich, #252549, Saint-Quentin-Fallavier, France) in PBS. Cells were washed 3 times in PBS, 0.1% Tween-20 (100 µL per well), then primary anti-alpha-tubulin antibody (clone α3A1), 1:5000 in PBS 2% Bovine Serum Albumin (BSA), was added for 45 min. Cells were washed twice again and secondary anti-mouse antibody coupled to HRP (1:2000 in PBS 2% BSA, #715-035-150, Jackson Immuno-Research Laboratories, Cambridgeshire, UK) was added for 45 min. Then, cells were washed again with PBS and 100 µL of ECL substrate (#170-5061, Bio-Rad Laboratories Inc., Hercules, CA, USA) was injected in each well using the FLUOstar OPTIMA Microplate Reader (BMG Lab technology, Champagny-sur-Marne, France). The luminescent signal was read immediately after ECL injection. IC_50_ values were calculated using GraphPad Prism software (10.3.1 version) and are presented in the text as means ± SEM of three independent experiments.

#### 3.2.5. Immunofluorescence

Cells at a density of 20,000 cells per well were grown for 48 h on glass coverslips placed in a 24-well microplate. When cells reached 70% confluence, the medium was replaced with a fresh one supplemented with DMSO (0.005% or 0.01%) or the test compound at 1 or 200 µM. After 30 min of incubation, cells were permeabilized in warm OPT buffer (80 mmol/L Pipes, 1 mol/L EGTA, 1 mol/L MgCl_2_, 0.5% Triton X-100 and 10% Glycerol, pH 6.8) and fixed for 6 min in −20 °C methanol (Carlo ERBA SAS, #414855, Val-de-Reuil, France). After washing and saturation with a specific blocking buffer (3% BSA), 10% Goat serum (Gibco Invitrogen, #16210064, Carlsbad, CA, USA) in PBS, cells were incubated for 45 min at room temperature (RT) with anti-alpha-tubulin antibody (clone α3A1 in blocking buffer). Cells were washed twice again and subsequently incubated with Alexa 488 conjugated anti-mouse antibody (1:500 in blocking buffer, #115-545-166, Jackson immune-research laboratory, Cambridgeshire, UK) and DNA was stained with Dapi which was included in mounting media (Dapi fluoromount, #0100-20, Clinisciences, France). Images were captured with a Zeiss AxioimagerM2 microscope (Zeiss microscopy, Rueil Malmaison, France) equipped with the acquisition software AxioVision (4.8 version, Marly-le-Roi, France).

#### 3.2.6. Protein Extraction and Western Blot Analysis

For each cell line, cells were treated or not (in the case of the control) with three concentrations of chalcones **10** and **13** (according to IC_50_ values at 48 h) and then harvested with trypsin. For total protein extraction, collected samples of each condition were washed in PBS. Then, the total cell pool was centrifuged at 200 g for 5 min at 4 °C and homogenized in RIPA lysis buffer (50 mM HEPES, pH 7.5, 150 mM NaCl, 1% sodium deoxycholate, 1% NP-40, 0.1% SDS, 20 mg/mL of aprotinin) containing protease inhibitors according to the manufacturer’s instructions. After quantification of protein levels using the Bradford method, proteins (60 mg) were separated on 12.5% SDS-PAGE gels and transferred to PVDF membranes (GE Healthcare Life Science—Fisher Scientific). Membranes were probed with respective antibodies against human caspase-3, cleaved caspase-3, and PARP according to the manufacturer’s instructions. After incubation with appropriate secondary antibodies, blots were developed using the “ImmobilonWestern” substrate, following the manufacturer’s protocol, and G:BOX system (Syngene, Cambridge, UK). Membranes were then reblotted with human anti-β-actin used as a loading control.

## 4. Conclusions

Aiming to design new anticancer lead molecules and to complete our structure–activity relationship studies, fifteen chalcones were synthesized and evaluated for their antiproliferative activity in human CRC HCT116 and HT-29 cells and in human PCa cancer DU145 and PC3 cells. Most of the synthesized compounds displayed IC_50_ values lower than 5 µM at 48 h for the HT-29 cell line, which is known for its greatest anticancer drug resistance. In the phenolic series, chalcone **8**, a combretastatin-A4 (**C-A4**) analogue, was found to be the most potent compound. The 3-hydroxy group was identified as an important structural feature since chalcone **2** also revealed interesting IC_50_ values, especially against CRC cells. This antiproliferative effect seemed to be in correlation with the inhibitory activity on cell microtubules; chalcone 8 is well-known for such an inhibitory effect on tubulin polymerization but this effect is demonstrated for the first time for chalcone **2**. A further study will be devoted to the vectorization of these phenolic chalcones by active targeting, through the coupling to a polyamine chain in order to increase the selectivity towards tumor cells.

Concerning heterocyclic chalcones, we confirmed the potency of the indolyl chalcone as scaffold for designing anticancer agents. Thus, chalcone **10** displayed IC_50_ values lower than 40 nM at 48 h and will also be subjected to vectorization, as mentioned above. Finally, a new series of chromonyl chalcones has been designed, synthesized in good yields, and highlighted for its biological interest, especially chalcone **13**. The latter, as well as chalcone **10**, were therefore further investigated for their mechanism of action; their marked in vitro antiproliferative effect was linked to apoptosis induction through several pro-apoptotic markers: activation of caspase-3 and PARP cleavage. Thus, chalcone **13** is selected for a strategy of vectorization by passive targeting, as described for previous chalcones included into β-cyclodextrins/ cellulose nanocrystals (CNCs) complexes [28]. Indeed, the enhancement of the hydrosolubility of this hydrophobic chalcone could result from the encapsulation into the lipophilic cavity of β-cyclodextrin, which is associated with hydrophilic cellulose nanocrystals. Furthermore, CNCs would serve as vectors to specifically target tumor cells through the enhanced permeation and retention effect.

## Data Availability

The original contributions presented in the study are included in the article/Appendix A, further inquiries can be directed to the corresponding author.

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
