# Peer review of "Synthesis and Antiproliferative Effect of 3,4,5-Trimethoxylated Chalcones on Colorectal and Prostatic Cancer Cells"

_pharmaceuticals, 2024, doi:10.3390/ph17091207_

Round 1

Reviewer 1 Report (Previous Reviewer 1)

Comments and Suggestions for Authors

The revised version of the MS entitled "Synthesis and antiproliferative effect of 3,4,5-trimethoxylated chalcones on colorectal and prostatic cancer cells" is greatly improved. The authors have addressed all of my concerns.

Author Response

Thank you for these comments.

Reviewer 2 Report (New Reviewer)

Comments and Suggestions for Authors

The manuscript describes the synthesis of 15 chalcone derivatives and evaluation of their anticancer activity.  The authors got 4 compounds with promising activity profile. The authors have also carried out the SAR studies to get more insights about the superior activity and structural specificity of potent molecules. The references are found to be adequate and the obtained results clearly underlines the importance of the work carried out. This work will be a useful contribution to the scientific community, especially for researchers working in the area of drug-discovery.

This work is interesting and can be published in "Pharmaceuticals" after minor revision. Some of the recommendations are detailed below:

Major suggestions

1) In the experimental section, the mmol of reagents used should be mentioned to understand whether the reaction was carried out in gram scale or milligram scale.

2) Pages 17,18, Synthesis of 1,3,7 and 9: Is it 25 equivalents of base? Reassure it during revision.

Minor suggestions

1) There are typos and grammatical errors that needs to be addressed.

1.a) Line 96: "for example [22]" should be rephrased.

1.b) Lines 120, 421, 536 and scheme 3: Correct to  "piperidine".

1.c) Line 49: Rephrase "doublet both".

1.d) Line 297: change "on" to "in".

1.e) Lines 419 and 420: In heading, it is "compounds 1-10" and in next line it is "compounds 1-11". 

1.f) Change "eq." to "equiv." wherever applicable.

1.g) Line 700: Correct to "(BSA)" instead of "(BSA))".

1.h) Line 726: Rephrase "were treated or not with". It is not clear.

Comments on the Quality of English Language

Minor editing is needed.

Author Response

This manuscript is a resubmission of an earlier submission. The following is a list of the peer review reports and author responses from that submission.

Round 1

Reviewer 1 Report

Comments and Suggestions for Authors

In the MS entitled “Synthesis and antiproliferative effect of 3,4,5-trimethoxylated chalcones on colorectal and prostatic cancer cells”, authors synthesized 16 chalcones bearing a 3,4,5-trimethoxylated A ring and a variety of B rings and tested their antiproliferative/cytotoxic effects against colorectal and prostate cancer cells. Chalcone (4) demonstrated excellent cytotoxicity, while the novel chromonyl chalcone (7) showed promising antiproliferative activity. Further investigations into their mechanisms of action are planned for future studies. Although interesting, a few issues need to be addressed prior to publication, especially regarding the interpretation of the MTT assay, and the lack of another experiment to validate the MTT data. Please find a list of issues that should be addressed:
• I couldn’t understand this sentence: “Besides, prostate cancer is the most commonly diagnosed cancer in men even if its mortality is lower than that of previous malignant tumors”.
• What is the rationale for testing colorectal cancer and prostate cancer cell lines?
• Please describe how many cells were plated per well. The concentration of cells per mL without the volume per well does not clarify the cell count per well.
• It is unclear if the MTT assay was designed as a cytotoxicity or proliferation assay. Another parallel assay is needed to validate MTT data. Please clearly define the MTT assay design and why it addresses cytotoxicity vs proliferation. A second assay should be performed to validate either the cytotoxicity or the proliferation insights.
• MTT assay measures cell viability by assessing cellular metabolic activity. Can the author verify the findings using another cell viability assay in parallel? Not every concentration has to be tested. The parallel assay experiment is important to validate the MTT assay data.

Reviewer 2 Report

Comments and Suggestions for Authors

This manuscript presented the synthesis of new chalcone derivatives and evaluation of anticancer activity. As a result, by one step reaction, authors obtained effective cytotoxic agents acting in the nanomolar range. The positive effect of the indole ring has been repeatedly noted in various areas of medicinal chemistry, and this work confirms existing data. This is very good results, especially for scientists working in this area. From my side, there are no critical notes for fixing the article. As a small request, for better perception, it will be nice if authors will add into the Fig.2 the structure of 3-hydroxy-3’,4,4’,5’-tetra-methoxychalcone near the combretastatin-A4. 

Comments on the Quality of English Language

Minor editing of English language required

Author Response

As a small request, for better perception, it will be nice if authors will add into the Fig.2 the structure of 3-hydroxy-3’,4,4’,5’-tetramethoxychalcone near the combretastatin-A4. 

Reply: the structure of the 3-hydroxy-3’,4,4’,5’-tetramethoxychalcone has been added next to that of combretastatin-A4 into the Fig.2. 

Reviewer 3 Report

Comments and Suggestions for Authors

In this manuscript, Letulle et al., described the synthesis and anticancer activity of trimethoxy phenyl based chalcone derivatives. The authors also proposed a SAR study based on nine compounds and other additional three analogs.

The manuscript lacks novelty and significant. The potent compound 4 claimed in this manuscript was already reported in the literature by Kumar et al. I couldn't find any novel compounds that were promising or noteworthy. Furthermore, the cytotoxicity assay is the only biological activity addressed in this publication. The SAR studies proved inconclusive due to the small number of compounds used. The discussion and conclusion section were poorly written.

This manuscript did not meet the quality standards of the Pharmaceuticals journal.  Therefore, I would not recommend the publication of this manuscript in Pharmaceuticals. 

Round 2

Reviewer 1 Report

Comments and Suggestions for Authors

The revised version of the MS is much improved. The authors improved their findings by performing another assay: the Trypan Blue Dye Exclusion method. It is unclear why the authors would not include such an assay in the MS. The Trypan Blue Dye assay suggests that the effect of Compound 4 is due to reduced proliferation and that Compound 7 has an additional impact by inducing toxicity, depending on the cell type. I believe the MS would be much improved if the data was incorporated into the final version of the MS. Also, MTT at times can show reduced “proliferation” or “viability”, but the effect of the compound could be a direct effect on cellular respiration or in affecting the reduction of MTT into formazan. Therefore, this additional experiment validates the  MS findings. Please incorporate the data into the MS or clarify why this data is not incorporated into the MS (Is it going to be implemented in future studies?).

Reviewer 3 Report

Comments and Suggestions for Authors

In this manuscript, Letulle et al synthesized and tested a few 3,4,5-trimethoxylated chalcones in colorectal and prostate cancer cells. Here are a few recent articles from Pharmaceuticals. The majority of recent studies involved extensive synthetic efforts, as well as biological activity and mechanistic investigation.

Pharmaceuticals 202316(10), 1354 (https://doi.org/10.3390/ph16101354) explain the synthesis, SAR, anticancer activity, ADMET properties and the follow up mechanistic studies along with docking analysis. More than 30 chalcones were synthesized and tested in 3 different cancer cell lines with proper SAR alanysis.

Pharmaceuticals 202316(9), 1331; https://doi.org/10.3390/ph16091331 explains the synthesis, SAR, biological activity and the mechanistic studies

Pharmaceuticals 202316(6), 879; https://doi.org/10.3390/ph16060879 explains the novel synthesis of novel chalcones, mechanistic studies and anticancer properties

Pharmaceuticals 202316(1), 83; https://doi.org/10.3390/ph16010083 explains the synthesis, docking studies, mechanistic analysis and the biological activity.

Pharmaceuticals 202215(3), 280; https://doi.org/10.3390/ph15030280 explains the synthesis, docking studies, mechanistic analysis and anticancer activity

In addition, the review article titled “Chalcones: Promising therapeutic agents targeting key players and signaling pathways regulating the hallmarks of cancer” lists important chalcones with anticancer activities. Given the importance of chalcones in the anticancer arena, without the proper mechanistic studies, I will not recommend the article for publication. However, I advise the authors to carry out the mechanistic studies to be suitable for prestigious journals such as Pharmaceuticals. These are some of the outstanding questions,

1.      What is the reason for selecting colon cancer and prostate cancer for this study?

2.      The potent chalcone compound listed in this manuscript 4 is already synthesized and available in the literature. However, the authors found that this compound had a promising <50 nM IC50 against the cell lines listed here. What is the mechanism behind this promising anticancer activity?

3.      Figure 4-6 are the repetition of Tables 1-4. This should be moved to supplementary information.
